# Enhanced Heat Resistance in *Morchella eximia* by Atmospheric and Room Temperature Plasma

**DOI:** 10.3390/microorganisms12030518

**Published:** 2024-03-05

**Authors:** Qin Zhang, Junbin Lin, Junjie Yan, Renyun Miao, Rencai Feng, Ying Gan, Bingcheng Gan

**Affiliations:** 1Institute of Urban Agriculture, Chinese Academy of Agricultural Sciences, Chengdu 610299, China; 2Chengdu National Agricultural Science and Technology Center, Chengdu 610299, China

**Keywords:** Mel-7, ARTP mutagenesis, heat resistance, glutathione, antioxidant enzyme activity

## Abstract

This study focuses on optimizing the mutagenesis process for *Morchella eximia* (Mel-7) mycelia through atmospheric and room temperature plasma (ARTP) mutation and explores the resultant thermal adaptability and physiological responses of mutant strains. This research demonstrated a clear relationship between ARTP mutagenesis exposure duration and lethality rate, indicating that an exposure time of 40 s resulted in the optimal balance of inducing mutations without causing excessive mortality. Additionally, this study established 43 °C as the ideal screening temperature for identifying mutant strains with enhanced heat resistance, as this temperature significantly challenges the mycelia while allowing thermotolerant strains to be distinguishable. Among the screened mutants, strains L21, L23, L44, and L47 exhibited superior growth and high-temperature tolerance, with notable resilience at 30 °C, highlighting their enhanced adaptability to above-optimal temperatures. Furthermore, this research delved into biochemical responses, including lipid peroxidation and non-enzymatic antioxidant content, highlighting the diverse mechanisms, such as enhanced lipid peroxidation resistance and increased antioxidant content, employed by mutant strains to adapt to temperature fluctuations. The activities of antioxidant enzymes, including peroxidase (POD) and superoxide dismutase (SOD), were shown to be significantly influenced by temperature elevations, illustrating their critical roles in the thermal adaptation of mutant strains. These findings shed light on the importance of considering mutation duration and temperature screening in the development of thermotolerant fungal strains with potential applications in various industries. This study’s breakthrough lies in its comprehensive understanding of the thermal adaptability of Mel-7 mycelia and the identification of promising mutant strains, offering valuable insights for both academic and industrial purposes.

## 1. Introduction

Morel mushrooms (*Morchella* spp.) exhibit a variable sensitivity to high-temperature environments, with this sensitivity affecting their growth to varying degrees across species, leading to consistent impacts on yield and quality [1]. This susceptibility presents a significant challenge to their industrial utilization and cultivation. Given their comprehensive nutritional profile, which includes proteins, amino acids, vitamins, and trace elements, and their diverse applications across the food seasoning sector (such as flavor enhancers in gourmet dishes), health product formulation (e.g., dietary supplements), and pharmaceutical development (including acting as sources of bioactive compounds), enhancing the thermal resilience of morels is crucial [2,3,4,5]. Therefore, focused research efforts on genetic and environmental strategies to strengthen their heat tolerance and improve their ability to withstand elevated temperatures are essential. Such research is vital for their survival and key to optimizing their commercial potential and ensuring the sustainable growth of the industries they support.

*Morchella*, commonly known as true morels, are fungi belonging to the *Ascomycota* division, specifically within the *Pezizomycotina* subphylum, showcasing their unique taxonomic position [6]. These fungi are distinguished by their distinctive honeycomb-like appearance, featuring caps with a complex network of ridges and pits. Morels are found worldwide, including in diverse regions of China such as Sichuan, Yunnan, Gansu, Henan, Hebei, Hunan, Heilongjiang, Xinjiang, and Jiangsu, highlighting their extensive geographical distribution [7,8]. Morels, often celebrated as the “King of Mushrooms”, are highly valued for their luxurious culinary uses. Analytical studies have shown that 100 g of dried morels contain high concentrations of potassium and phosphorus, seven and four times higher, respectively, than those found in Cordyceps sinensis [9]. Moreover, their zinc and iron content significantly surpass that of shiitake mushrooms and *Hericium erinaceus*, making them an exceptional nutritional choice, especially for vegetarians. This outstanding nutritional profile firmly establishes morels as a delicacy for vegetarians. Beyond their culinary appeal, morels are recognized for their medicinal properties, attributed to bioactive compounds like polysaccharides, phenolic compounds, and terpenoids. These components endow morels with antioxidant, anti-inflammatory, immunomodulatory, and antimicrobial activities, making them of significant interest to the pharmaceutical and health product industries for the development of innovative therapeutic agents and wellness solutions [4]. Consequently, the pharmaceutical and health product sectors have shown substantial interest in harnessing morels for the development of innovative therapeutic agents and wellness solutions.

The mycelial growth of *Morchella* spp. is susceptible to temperature, necessitating precise regulation for optimal development. Experimental findings [10] have shown that while temperatures exceeding 25 °C can indeed accelerate mycelial growth, these conditions may not foster the desired development, characterized by dense, efficient nutrient-transporting mycelial networks essential for the formation of high-quality fruiting bodies. Rapid proliferation under high temperatures can lead to the formation of loosely interconnected mycelial structures, undermining the structural integrity vital for nutrient distribution and robust colony growth. This can result in sparse and inconsistent colony development, adversely affecting yield and quality. Moreover, elevated temperatures may induce accelerated mycelial aging, manifested by increased metabolic activity that reduces the lifespan and vitality of mycelia, further diminishing yield and quality. To mitigate these adverse effects, meticulous temperature management, employing techniques such as indoor cultivation or using shade cloths and irrigation in outdoor settings, is paramount in morel cultivation. Keeping temperatures within the ideal range of 18–22 °C ensures controlled mycelial growth and overall health [11]. Temperature thus plays a pivotal role in the growth dynamics of Morel’s mycelia, with improper thermal conditions leading to undesirable growth patterns and premature aging. Precise temperature control is essential for achieving healthy mycelial growth and superior yields. With the increasing prevalence of extreme climatic events, the agricultural stability of morels, traditionally cultivated in open fields, is at risk. Pursuing strains with resilience to elevated temperatures, through genetic selection or breeding programs, offers a promising strategy to combat the challenges posed by temperature volatility, ensuring the sustainable production of morels.

Atmospheric and room temperature plasma (ARTP) has emerged as an advanced plasma mutagenesis technology, which is particularly effective in generating biological mutants, including significant advancements in the breeding of edible fungi. Since its inception in 2011, ARTP has rapidly expanded its scope to include the breeding of yeast and bacterial strains, with a strategic focus on enhancing their overall performance and yield, notably in yeast and bacterial strains, and especially in improving the productivity and quality of edible fungi [12,13]. This technological trajectory has evolved further as ARTP has been applied to a diverse array of microorganisms, including a wide range of microorganisms with a special emphasis on edible fungi, alongside bacteria, yeasts, and algae, resulting in the development of various effective breeding techniques [14]. During the 2020s, the application spectrum of ARTP was extended even further, encompassing animal and plant breeding as well as biomedical research, thus establishing a novel multidisciplinary platform [15,16]. The underlying principle of ARTP involves the use of ionized gas at ambient conditions to induce mutations. This plasma, generated at atmospheric pressure, has the capability to cause DNA damage, leading to genetic variations. The unique attributes of ARTP, such as high mutation efficiency, an extensive mutation spectrum, and minimal detrimental impact on cell viability, make it particularly suitable for the targeted genetic enhancement of edible fungi. Recently, ARTP’s applicability in enhancing edible fungi has been demonstrated through significant improvements in yield, nutritional composition, and disease resistance, showcasing its potential for groundbreaking advancements in this field [17,18,19,20]. The evolution and deployment of ARTP mutation technology represent a significant advancement in the field of mutation breeding. With a widening range of applications, this cutting-edge technology promises a promising future, poised to make substantial contributions to various scientific disciplines.

In the pursuit of securing a desirable mutant strain with enhanced thermotolerance—the ability to withstand higher temperatures—the careful selection of specific mutation techniques (such as chemical mutagenesis and UV irradiation) and screening methods is of utmost importance. The process involves exposing microorganisms to nearly lethal temperatures for a specified duration, followed by a meticulously controlled cultivation phase. Surviving organisms from this heat stress are subjected to a rigorous multi-stage screening process under various experimental conditions, aimed at isolating thermotolerant strains.

In this study targeting high-temperature-sensitive strains of *Morchella eximia* (Mel-7), we employed ARTP technology to facilitate targeted mutations. After treatment, the mycelia underwent a precise screening regimen at 43 °C, specifically designed to identify strains capable of thriving at elevated temperatures. Our methodology was extended to include comparative analyses conducted at different temperature thresholds, along with quantification of antioxidant activity and lipid peroxidation levels. This comprehensive approach enabled the accurate identification and validation of thermotolerant strains, providing valuable insights into the manipulation of genetic traits within *Morchella eximia*, paving the way for the development of commercially viable strains with enhanced resilience to climate change. The utilization of ARTP in this context underscores its effectiveness as a potent tool for genetic enhancement, offering significant potential for further exploration in the field of edible fungi.

## 2. Materia and Methods

### 2.1. Materials

The *Morchella eximia* used in the experiment, denoted as Mel-7 [21], was provided by the Institute of Urban Agriculture, Chinese Academy of Agricultural Sciences. The cultivation conditions involved incubation at 20 °C while completely avoiding light exposure.

In each inoculation, a tool with an 8 mm diameter puncher was used to create a hole at the edge of the colony. Subsequently, mycelial plugs were transferred to a sterile Potato Dextrose Agar (PDA) medium using a sterile inoculation needle in an aseptic manner and then incubated at 20 °C.

### 2.2. Determination of Lethality Rates

The Mel-7 strain, initially stored at 4 °C, underwent a triple reactivation process on PDA medium. The reactivated mycelia were then placed into a sterilized homogenizer (500 r/s, HH-200C, Yiliantang, Zhejiang, China), to which 100 mL of potato dextrose broth (PDB) was added. The mycelia were homogenized for 2 min to achieve a uniform suspension of individual or multiple free mycelial fragments.

ARTP mutagenesis (ARTPIII, THU-G-2014048, Si Qing Yuan Bio-technology Co., Ltd., Beijing, China) was performed using helium gas, with a power setting of 120 W and a distance of 2 mm between the workbench and the gas outlet, employing a standard nozzle for uniform gas distribution. The samples were exposed to ARTP for varying durations, ranging from 10 to 70 s, selected based on preliminary dose–response studies.

The mutagenized mycelia were then diluted appropriately, evenly spread onto PDA medium, and cultivated at 20 °C for 2–3 days. After this period, individual colonies appeared. The colony count was recorded, and lethality rates were calculated using the formula: (1 − (number of colonies post-treatment/number of colonies pre-treatment)) × 100%.

### 2.3. Mutagenesis and Screening

The optimal mutagenesis duration for this experiment, aiming for a target lethality rate of approximately 90%, and the subsequent screening procedures were carried out as follows (Figure 1):

Using a T-shaped hole puncher designed to extract uniform agar blocks containing fungal mycelia, the blocks were transferred into a glass mortar containing PDB with the aid of an inoculation needle. The agar blocks were ground to achieve a homogeneous slurry, ensuring uniform exposure during mutagenesis. Using a pipette, 10 μL of this slurry was placed on a metal sheet, which was then positioned under the ARTP mutagenesis device for a 40 s treatment, a duration determined to be optimal based on preliminary dose–response studies. After mutagenesis, the metal sheet was designed to automatically fall into a centrifuge tube containing sterile water, a feature facilitated by the ARTP device’s integrated mechanism. The process continued with the resuspension and dilution of the fungal liquid, from which 100–150 μL was spread onto the PDA medium. The plates were incubated at 20 °C for 2–4 days until visible fungal colonies formed. These colonies underwent a 90-minute heat shock treatment at 43 °C to select for specific traits, followed by further incubation at 20 °C for 5–7 days. Vigorously growing mycelial segments, selected based on their robust growth and morphology, were then transferred to a new PDA medium, marking the establishment of the M1 generation.

### 2.4. Mycelial Growth Rate Determination

Using a sterile 8 mm diameter puncher, holes were made at the edge of the mycelial colony on active plates. The mycelia extracted from each hole were then inoculated onto PDA medium and incubated at 20 °C in complete darkness. To ensure reproducibility, each treatment was conducted with three replicates. The radial growth of the mycelia were assessed using the cross method [22], which involves scoring the plates with a cross immediately after inoculation. Upon initial growth, the longest radius from the center to the tip of the growth (r1) was marked. Two days later, a second measurement (r2) was made at the furthest extent of the mycelial growth. The distance between r1 and r2 was measured using a vernier caliper (in mm) and divided by the number of days (d) the mycelia were cultured to calculate the average linear growth rate per day of the mycelia. This procedure ensured the precise tracking of mycelial expansion over time.

### 2.5. High-Temperature Resistance Test

Mutant strains identified through a 43 °C high-temperature screening process were subjected to sequential subculturing across 3–4 generations at 20 °C to evaluate the stability and heritability of their phenotypic traits. Throughout this phase, those mutants exhibiting mycelial growth and vitality either comparable to or surpassing the non-mutagenized Mel-7 strain were selected for further analysis, with the Mel-7 strain serving as the wild-type control (WT).

In the fourth generation (M4), these selected mutants were tested for high-temperature tolerance using a resistance assay at 37 °C. The fifteen identified mutant strains, along with the WT, were initially cultivated at 20 °C for two days. Subsequent to this period, their growth metrics were documented, followed by exposure to a 37 °C environment for 48 h, and then a week-long cultivation at 20 °C. Observations and measurements of mycelial growth were conducted to isolate strains that exhibited enhanced high-temperature resilience.

Upon evaluating the strains at 37 °C, four mutant strains demonstrated significant tolerance to high temperatures. To further assess the durability of this trait, these four strains, alongside the WT strain, were subjected to four cycles of growth at 30 °C, with each cycle spanning three days. The growth rate and morphological changes were meticulously documented through photographic records and quantitative measurements to track their adaptation over time.

### 2.6. Activity of SOD and POD

The selected strains exhibiting high-temperature resistance (M7), along with the WT, underwent a standardized procedure using an 8 mm punch to extract uniform mycelial plugs from the periphery of the colonies. These plugs were meticulously transferred in a sterile manner onto Petri dishes, prepared with sterile filter paper and PDA medium, with 12 plates allocated for each strain. The distribution of the plates was as follows: three plates were incubated at 20 °C for 6 days to serve as a control, three plates were incubated at 20 °C for 5 days before undergoing heat stress at 37 °C for 24 h to assess heat shock response, and the remaining six plates were incubated at 30 °C for 6 days to evaluate sustained temperature tolerance. Upon completion of the incubation periods, the mycelia grown on the surface of the filter paper was delicately scraped off using a sterile scalpel and collected into 2.0 mL centrifuge tubes, and their dry weights were recorded to ensure uniformity in measurement. Following previously described protocols for sample preparation, which included drying and homogenization, the enzymatic activity within the mycelia was measured. This involved the use of SOD (Catalog Number: AKAO001M) and POD (Catalog Number: AKAO005M) assay kits from BoxBio (Beijing Boxbio Science & Technology Co., Ltd., Beijing, China), strictly adhering to the detailed experimental guidelines provided in the instruction manual. The selection of SOD and POD assays was based on their relevance in assessing the oxidative stress response, a critical factor in high-temperature tolerance.

### 2.7. Measurement of Lipid Peroxidation

The level of lipid peroxidation in biological organisms is determined by measuring malondialdehyde (MDA), a decomposition product of peroxidized polyunsaturated fatty acids in the membrane. MDA reacts with thiobarbituric acid (TBA) to form a reddish-brown compound, 3,5,5-trimethyl-2,4-dioxazolidine, under specific acidic conditions and high temperatures. This compound exhibits maximum absorbance at a wavelength of 532 nm, allowing for the estimation of peroxidized lipid content in the sample through colorimetry at this wavelength [23].

However, when measuring MDA levels in animal and plant tissues, various substances, notably soluble sugars, can interfere. These sugars can react with TBA, leading to maximum absorbance at 450 nm, albeit with some absorbance at 532 nm as well. To mitigate these interferences, absorbance values are measured at 600, 532, and 450 nm. The malondialdehyde content is quantitatively determined by calculating the difference in absorbance values at these wavelengths, following a specified formula or method.

Sample pretreatment was conducted as outlined in Section 2.6. The MDA content in the mycelia was determined using a malondialdehyde content detection kit (Catalog Number: AKFA013M) from BoxBio (Beijing Boxbio Science & Technology Co., Ltd., Beijing, China), with measurements and analysis performed according to the kit’s provided instructions, which are based on the TBA-MDA reaction principle.

### 2.8. Detection of GSH and GSSG

Glutathione, a tripeptide composed of γ-glutamyl, cysteinyl, and glycyl residues (glutamic acid, cysteine, and glycine), plays a pivotal role in cellular processes including antioxidant defense and amino acid transport. It exists in two forms: reduced glutathione (GSH) and oxidized glutathione (GSSG), with GSH being the predominant intracellular thiol compound. The GSH to GSSG ratio is a critical indicator of the cellular redox state [24].

Following the pretreatment protocols outlined in Section 2.6, the levels of GSH and GSSG in mycelia were assessed. This was achieved using the reduced glutathione content detection kit (Catalog Number: AKPR008M) and the oxidized glutathione content detection kit (Catalog Number: AKPR009M) from BoxBio, with procedures adhering to the kits’ instructions. These kits employ a specialized measurement technique to accurately quantify the levels of both reduced and oxidized glutathione.

### 2.9. Statistical Analysis

The experiments were conducted in triplicate, with results presented as the mean ± standard deviation of these values. Data analysis was performed using analysis of variance (ANOVA), followed by Tukey’s multiple comparison test, to identify statistically significant differences among the groups. This analysis was carried out using DPS data processing software (Version 9.01) [25]. Statistical significance was assigned to differences with a *p*-value less than 0.05.

## 3. Results

### 3.1. Determining the Best Mutagenesis Duration and Optimal Screening Salinity

The effectiveness of ARTP mutation breeding significantly depends on accurately calibrating the mutation dosage, defined as the intensity and duration of exposure to mutagenic conditions [26]. As demonstrated in our results (Figure 2A), there was a clear correlation between ARTP mutagenesis exposure time and the lethality rate in Mel-7 mycelia, with 30, 40, and 50 s of exposure leading to lethality rates of 68.35%, 92.57%, and a complete 100%, respectively. Given the established model suggesting that a lethality rate above 90% increases the chance of obtaining beneficial mutations [27], a 40 s exposure was deemed optimal for further studies.

To determine the ideal screening temperature for mutation selection, we conducted a temperature sensitivity test. After a two-day culture period, the mycelia were exposed to heat ranging from 37 °C to 47 °C for 1.5 h, followed by a recovery period at 20 °C for a week (Figure 2B). Observations, including growth rate measurements and visual assessments of vitality, indicated that mycelium vitality was maintained up to 41 °C, beyond which a significant decline was noted.

To identify the optimal duration of heat shock that Mel-7 mycelia could withstand without compromising viability, we exposed the mycelia to 43 °C for 1, 1.5, and 2 h (Figure 2C). A one-hour treatment was observed to support continued growth and vitality, while longer exposures led to a significant reduction in vitality, pinpointing a 1.5-hour exposure as the threshold for maximal stress without detrimental effects on growth.

### 3.2. Temperature-Based Screening of Mel-7 Mutant Strains

In this study, mutant strains were initially subjected to a screening process at a temperature of 43 °C to identify those capable of withstanding this level of heat stress. Following this, a proliferation of Mel-7 colonies was observed, from which 108 robust colonies were selected and systematically numbered for further analysis. After discounting strains with suboptimal growth, 15 promising mutant strains were identified. These strains demonstrated growth activity at 20 °C comparable to the WT, as quantitatively assessed by colony diameter measurement, as depicted in Figure 3A.

To isolate mutants with superior heat resistance, a subsequent high-temperature test was conducted at 37 °C, aiming to assess resilience under heat stress, as illustrated in Figure 3B. After a 48-hour exposure, whereas the WT strain showed signs of growth stagnation, the mutants exhibited resilience, maintaining their growth trajectory upon transfer to a 20 °C environment for one week. This resilience, particularly in terms of recovery from heat stress, was found to correlate with biomass, a relationship highlighted in Figure 3B. Specifically, strains L21, L23, L44, and L47 displayed significantly greater biomass compared with their counterparts, leading to their selection for in-depth exploration.

### 3.3. Enhanced Heat Tolerance of Mutant Strains at 30 °C

During the cultivation process, we occasionally encountered temperatures around 30 °C, a condition that, while not extreme, significantly influences Mel-7 mycelium growth and vitality. To explore the phenotypic resilience of mutated strains at this temperature, we conducted a detailed analysis. As demonstrated in Figure 4A, at 30 °C, the growth rates (defined as the daily increase in colony diameter) of the mutated strains were significantly higher than that of the WT, with values of 12.91 ± 0.49, 13.24 ± 0.22, 12.91 ± 0.33, and 12.56 ± 0.21 mm/d, respectively, compared to the WT’s rate of 11.09 ± 0.24 mm/d.

Furthermore, we assessed the continuous growth performance of the mycelia across four 3-day generations at this temperature (Figure 4B). The results showed a progressive decline in growth vitality across generations for all strains, yet the mutated strains maintained higher growth rates than the WT. Specifically, by the second generation, the mutated strains’ growth rates were consistent with their initial generation, whereas the WT exhibited a notable decrease. By the third generation, the mutated strains showed a reduction in growth, while the WT’s growth was severely compromised. Statistical analyses confirmed the significance of these differences, suggesting that mutated strains possess enhanced tolerance to the elevated temperature of 30 °C. This tolerance could have practical implications for the cultivation and genetic improvement of Mel-7 strains under varying environmental conditions.

### 3.4. Lipid Peroxidation

This study meticulously examined the oxidative stress response and antioxidant mechanisms in WT and mutant strains under varying temperatures. Lipid peroxidation, assessed by measuring malondialdehyde (MDA) levels, showed that at 20 °C, MDA content in the WT strain was 487.02 ± 11.22 nmol/L (Figure 5). Notably, mutant strain L21 exhibited a significantly higher MDA level (798.69 ± 21.01 nmol/L) than the WT, suggesting enhanced oxidative stress. Conversely, strain L47 displayed a lower MDA content (390.34 ± 14.26 nmol/L), indicating reduced lipid peroxidation.

Upon increasing the temperature to 30 °C, a distinct reduction in MDA content was observed in strains L21, L23, and WT, while strains L44 and L47 showed increases in MDA levels, suggesting variable thermal stress responses among the strains. After a 24-h heat shock at 37 °C, a pronounced decrease in MDA content highlighted the dynamic antioxidative adjustments across strains.

### 3.5. Non-Enzymatic Antioxidants

Glutathione (GSH), a critical intracellular antioxidant, plays a key role in mitigating oxidative damage through the scavenging of excess reactive oxygen species generated during cellular metabolism. At 20 °C, GSH content in the WT strain was measured at 30.51 ± 3.02 µg/g (Figure 6A). In contrast, mutant strains L21, L23, L44, and L47 demonstrated significant increases in GSH content, with respective increments of 20.21, 10.17, 32.57, and 22.81 µg/g, indicating enhanced antioxidative capacity. However, at 30 °C, the GSH content in the WT strain saw a marked decrease to 0.21 ± 0.08 µg/g, whereas the mutant strains showed notable increases, suggesting an adaptive response to elevated temperatures.

Following a 24-hour exposure to 37 °C, the GSH content in the WT strain significantly decreased to 0.62 ± 0.25 µg/g, whereas mutant strains L21, L23, L44, and L47 exhibited increases, highlighting their resilience to heat-induced oxidative stress.

For GSSG, the WT strain maintained minimal levels at 20 °C (0.03 ± 0.01 µg/g) (Figure 6B). Mutant strains, however, showed elevated GSSG contents, indicating a dynamic oxidative stress response. Upon culturing at 30 °C, WT GSSG levels remained low, while mutant strains displayed varied responses, further elucidating their differential adaptation to temperature stress.

Significant changes in GSSG content after a 37 °C heat shock underscore the complex antioxidative mechanisms at play, particularly in mutant strains, which either increased or decreased their GSSG levels, suggesting alterations in their redox state and stress resilience.

### 3.6. Activities of Antioxidant Enzymes

Under 20 °C cultivation conditions, the peroxidase (POD) activity in the WT strain was relatively low, at 3.64 ± 0.34 U/g, indicating baseline enzymatic activity per gram of fungal biomass (Figure 7A). In contrast, mutant strains showed a significant increase in POD activity, with values of 16.14 ± 3.70, 44.38 ± 1.47, 239.01 ± 37.44, and 105.44 ± 9.48 U/g, respectively, suggesting enhanced oxidative stress responses. As the temperature increased to 30 °C, the POD activity in the WT strain remained low (2.01 ± 0.01 U/g), while mutant strains exhibited a marked rise in activity, indicating their adaptive mechanisms to moderate temperature stress. Following a 24-hour heat shock at 37 °C, POD activity in the WT strain slightly increased to 2.57 U/g but notably decreased in the mutant strains, reflecting a general suppression of antioxidative enzyme activities under severe heat stress.

Similarly, superoxide dismutase (SOD) activity in the WT strain was 17.89 ± 1.89 U/g at 20 °C (Figure 7B). Mutant strains L21 and L23 showed higher SOD activities (29.40 ± 1.86 and 25.74 ± 4.39 U/g, respectively), while L44 had lower activity (12.73 ± 2.94 U/g), and L47’s activity was comparable to the WT. Under 30 °C conditions, all strains experienced an increase in SOD activity, demonstrating an overall enhancement in cellular defenses against oxidative stress. However, after a 24-hour heat shock at 37 °C, SOD activities drastically reduced, underscoring the vulnerability of both the WT and mutant strains to extreme thermal stress.

These observations reveal the significant role of temperature in modulating antioxidative enzyme activities in fungal strains, with mutant strains showing both enhanced activities under moderate stress and significant reductions under severe stress. This pattern underscores the complex regulation of antioxidative responses in fungi, highlighting the potential for genetic adaptations to confer improved stress tolerance.

## 4. Discussion

The study of thermal adaptability in fungal strains, with a focus on the Mel-7 mycelia, has unveiled pivotal insights with implications for both academic research and potential industrial applications. Our research sheds light on the efficacy of ARTP mutation breeding in enhancing the heat resistance of the Mel-7 strain. We found that precise control over the duration of mutation exposure and careful monitoring of the lethality rates are essential for optimizing mutation effectiveness. Furthermore, our investigation into the physiological responses of these strains revealed significant differences in lipid peroxidation and non-enzymatic antioxidant levels between the WT and mutant strains, underscoring the complex balance between induced genetic modifications and their biochemical consequences.

As we delve deeper into the implications of our findings, it becomes essential to integrate these insights into the broader context of fungal biology. This includes assessing the feasibility of leveraging these genetically adapted strains in areas such as agricultural production, environmental bioremediation, and the pharmaceutical industry, where enhanced thermal tolerance could offer substantial benefits. Moreover, understanding the cellular mechanisms that underpin these temperature-related physiological adaptations will be crucial in harnessing the full potential of these mutations. By exploring both the practical applications and the biological mechanisms at play, our research contributes to a nuanced understanding of fungal adaptability and the innovative use of mutation breeding techniques.

### 4.1. Optimization of ARTP Mutation Duration for Enhanced Trait Improvement

The efficacy of ARTP mutation, a method that utilizes plasma-induced mutagenesis to alter organism traits, is closely tied to the precise control over mutation duration. This control is essential due to its impact on the lethality rate, which significantly influences the diversity and viability of the resulting mutants. Our research supports targeting a lethality rate exceeding 90% as optimal for generating a broad spectrum of robust mutants, as this approach fosters the emergence of potentially advantageous genetic variations, corroborating the findings of Cao et al. (2017) [28].

Despite the advantages of a high lethality rate, it presents the challenge of possibly excluding valuable mutants that cannot endure such intense selection pressure. Moreover, mutation intensities beyond this threshold may impair mutant viability, echoing concerns raised in the literature [29,30,31].

Our investigations into the Mel-7 mycelia revealed that ARTP mutation profoundly affects survival and growth, with species-specific optimal mutation durations being crucial for desired outcomes. For Mel-7, a 40 s exposure emerged as ideal, balancing mutation induction with the preservation of microbial viability. This finding aligns with Liu et al. (2020) [32] and Yu et al. (2022) [33], highlighting the delicate trade-off between achieving beneficial mutations and avoiding detrimental genetic alterations.

Accordingly, while a lethality rate over 90% offers a promising strategy for trait enhancement through ARTP mutation, its application demands careful consideration of species-specific responses and experimental objectives, ensuring the selection of mutation durations that maximize potential benefits while minimizing adverse effects.

### 4.2. High-Temperature Tolerance of Mutant Strains

Our research demonstrated that mutant strains, particularly the Mel-7 mycelia, exhibited significant tolerance to high temperatures (30 °C), showcasing enhanced growth vitality compared with the WT. This enhanced tolerance, achieved through ARTP mutation, underscores the potential of these strains in applications requiring high-temperature resistance, such as in the agriculture and biotechnology sectors. Notably, the sustained growth performance of these mutant strains over four generations at 30 °C suggests their enhanced adaptation and practical utility in high-temperature environments.

The improved thermal adaptability of the mutated Mel-7 mycelia, with its capacity for robust growth up to 37 °C, supports theories that genetic modifications can bolster resilience against environmental stressors [34]. The observed growth decline at temperatures above 37 °C prompts a discussion on the potential causes, such as the denaturation of critical proteins and enzymes, and the limitations imposed by cellular membrane fluidity at elevated temperatures. These phenomena, possibly mitigated by specific mutations to some extent, highlight the complex interplay between genetic adaptations and biophysical limits, as discussed in the works of Russell et al. (2003) [35] and Guan et al. [36].

Comparative analysis with the WT strain revealed its relatively diminished performance under similar high-temperature conditions, illustrating the innate genomic constraints in coping with thermal stress. This comparison suggests that ARTP-induced mutations may target key genes associated with thermal tolerance, a notion echoed by Huang et al. (2018) in their study on thermotolerant yeast strains [37].

While ARTP mutation has significantly improved the Mel-7 mycelium’s thermal adaptability, it is evident that biological and biophysical constraints still define the upper limits of temperature tolerance. Future research could explore further genetic modifications or combine strategies to enhance the thermal resilience of fungal strains beyond these natural thresholds.

### 4.3. Differential Antioxidant Stress Response between Mutant and WT Strains

This study’s investigation into cellular lipid peroxidation and non-enzymatic antioxidant responses has shed light on the complex thermal tolerance mechanisms exhibited by different fungal strains. Malondialdehyde (MDA), a key marker of oxidative stress resulting from lipid peroxidation, varies significantly among strains under temperature fluctuations, indicating differing susceptibilities to oxidative damage [38]. For example, the increased lipid peroxidation in mutant strain L21 at 20 °C suggests potential challenges in maintaining cell membrane integrity, while strain L47’s lower MDA levels point to enhanced cellular resilience.

The observed decrease in MDA levels at 30 °C for strains L21, L23, and WT, contrary to expectations, raises questions about the permeability changes in cellular membranes at elevated temperatures and their effect on MDA efflux. Conversely, the increases in MDA levels for strains L44 and L47 at the same temperature underscore the unique responses of these strains to thermal stress.

Glutathione (GSH) and its oxidized form (GSSG) play pivotal roles in the cellular redox buffering system [24]. The significant rise in GSH content among mutant strains at 20 °C compared to the WT indicates a stronger antioxidative response. However, the variability in GSH and GSSG levels under elevated temperatures highlights the diverse strategies fungal strains employ to counteract oxidative stress.

The potential upregulation of antioxidative enzymes, such as superoxide dismutase (SOD) and catalase, in mutant strains suggests a multifaceted approach to enhancing thermal tolerance. This adaptation may contribute to the observed superior growth performance of mutant strains under high-temperature conditions, underscoring the critical role of an efficient antioxidant system in supporting fungal viability in thermally stressful environments [39].

In conclusion, the differential lipid peroxidation and antioxidant responses across fungal strains under varying temperatures underscore the complexity of their adaptation to thermal stress. Future investigations into the genetic and proteomic underpinnings of these responses are essential for unraveling the molecular basis of fungal thermotolerance, with a focus on identifying specific markers that confer enhanced resilience to high temperatures.

### 4.4. Antioxidant Enzymes in Temperature Adaptation and Oxidative Stress Mitigation

The adaptive response of organisms to fluctuating temperature conditions often hinged on their enzymatic defense mechanisms. Peroxidase (POD) and superoxide dismutase (SOD) are two pivotal antioxidant enzymes that played a significant role in mitigating oxidative stress, particularly in the context of temperature adaptation.

POD primarily acted on hydrogen peroxide (H_2_O_2_) and organic hydroperoxides, converting them into non-toxic compounds, thereby preventing lipid peroxidation and subsequent cellular damage [40]. Its activity is an indicator of a cell’s capability to cope with the overproduction of H_2_O_2_, especially during thermal stress. Elevated temperatures tended to amplify the rate of metabolic reactions, leading to enhanced production of reactive oxygen species (ROS), including H_2_O_2_. A responsive increase in POD activity might be indicative of an adaptive response to scavenge these excessive ROS.

SOD, on the other hand, catalyzed the dismutation of the superoxide anion (O_2_^−^) into either ordinary molecular oxygen (O_2_) or H_2_O_2_ [41]. It served as the first line of defense against superoxide radicals, which were often overproduced under stress conditions, including elevated temperatures. The function of SOD was complemented by POD since the H_2_O_2_ produced by SOD’s action was further detoxified by POD.

The observed sharp decline in the activities of these enzymes under high-temperature shocks could be attributed to several reasons. First, the denaturation of enzymes at elevated temperatures was a well-documented phenomenon [42]. The enzymes lost their tertiary structure, thereby becoming non-functional. Additionally, the high production rate of ROS under thermal stress might surpass the detoxification capacity of these enzymes, leading to their inactivation [43]. It is also conceivable that the rapid increase in temperature disrupted cellular homeostasis and membrane integrity, affecting the compartmentalization and function of these enzymes.

When mycelia are exposed to high-temperature environments, it leads to an increase in the concentration of H_2_O_2_ in the mycelial cells. The high levels of POD and SOD in the mutant strains can rapidly convert these peroxides into non-toxic compounds, preventing lipid peroxidation and subsequent cell damage. This ability allows the mycelia to survive in high-temperature environments for extended periods.

Understanding temperature-mediated modulation of antioxidant enzymes is crucial. Not only did these enzymes serve as markers of oxidative stress and cellular defense, but they also illuminated potential strategies for improving the thermo-tolerance of microbial strains, especially in the context of industrial applications.

## 5. Conclusions

This study utilized ARTP mutation technology to identify mutant strains capable of withstanding high temperatures and examined their adaptive mechanisms under such conditions. This breakthrough was invaluable for both academic research and real-world applications. The findings emphasized the importance of fine-tuning the ARTP mutation duration. Ideally, only about 10% of the organisms should survive the mutation process. This survival rate ensures a diverse selection of beneficial mutants while eliminating less robust strains. However, pushing this mortality rate too far could jeopardize the health of the remaining mutants. For the specific Mel-7 strain of mycelia, a mutation duration of 40 s proved optimal. These modified strains demonstrated an impressive ability to thrive in temperatures up to 30 °C, which had promising implications for industries such as agriculture and biotechnology seeking heat-enduring fungi. However, their performance declined beyond this temperature, indicating inherent biological boundaries. Interestingly, this study revealed that these mutant strains were more resilient to heat-related damages than their conventional counterparts, as evidenced by variations in lipid deterioration and innate protective measures. Recognizing the role of antioxidant enzymes, namely, POD and SOD, in counteracting this damage was essential. Such insights were pivotal for harnessing the potential of these strains, and future investigations should delve into their genetic structures to further understand their thermotolerance traits.

## Figures and Tables

**Figure 1 microorganisms-12-00518-f001:**
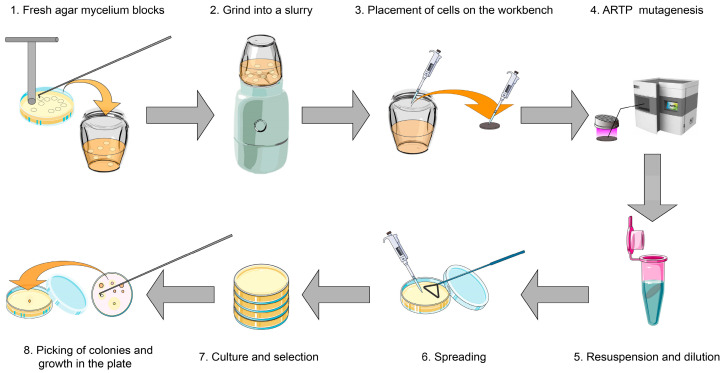
Atmospheric and Room Temperature Plasma mutagenesis workflow.

**Figure 2 microorganisms-12-00518-f002:**
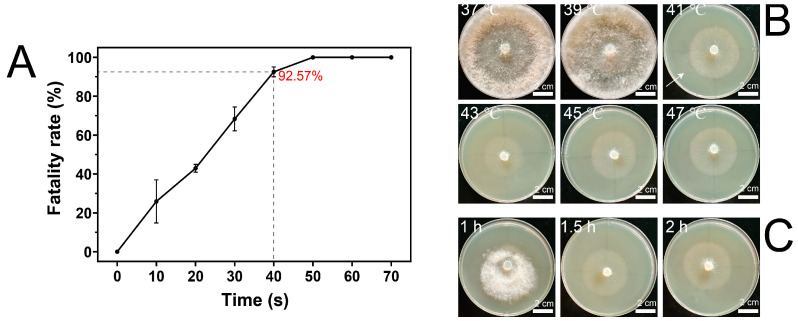
Lethality curve of Mel-7 and optimal screening temperature. The lethality curve of Mel-7 at various treatment times (**A**). The growth of Mel-7 mycelia was observed after heat stimulation at various temperatures for 1.5 h, followed by one week of cultivation at 20 °C (**B**). The white arrow indicates the mycelia newly grown at 20 °C for one week after being cultured at 41 °C for 1.5 hours. The growth of Mel-7 mycelia was examined after heat treatment at 43 °C for various durations, followed by one week of cultivation at 20 °C (**C**). The values presented in Figure 2C are the means ± standard deviations (SDs) of three independent experiments.

**Figure 3 microorganisms-12-00518-f003:**
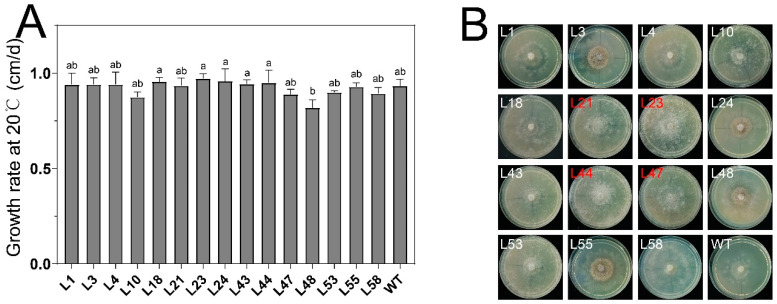
Mutant strains for high-temperature stress response. Growth rate of mycelium at 20 °C (**A**) and mycelium growth after 48 h of cultivation at 37 °C followed by one week of growth at 20 °C (**B**). Based on the consideration of mycelial biomass, L21, L23, L44, and L47 were chosen for further investigation. Values in a column with different letters are significantly different at *p* ≤ 0.05. The values presented are the means ± standard deviations (SDs) of three independent experiments.

**Figure 4 microorganisms-12-00518-f004:**
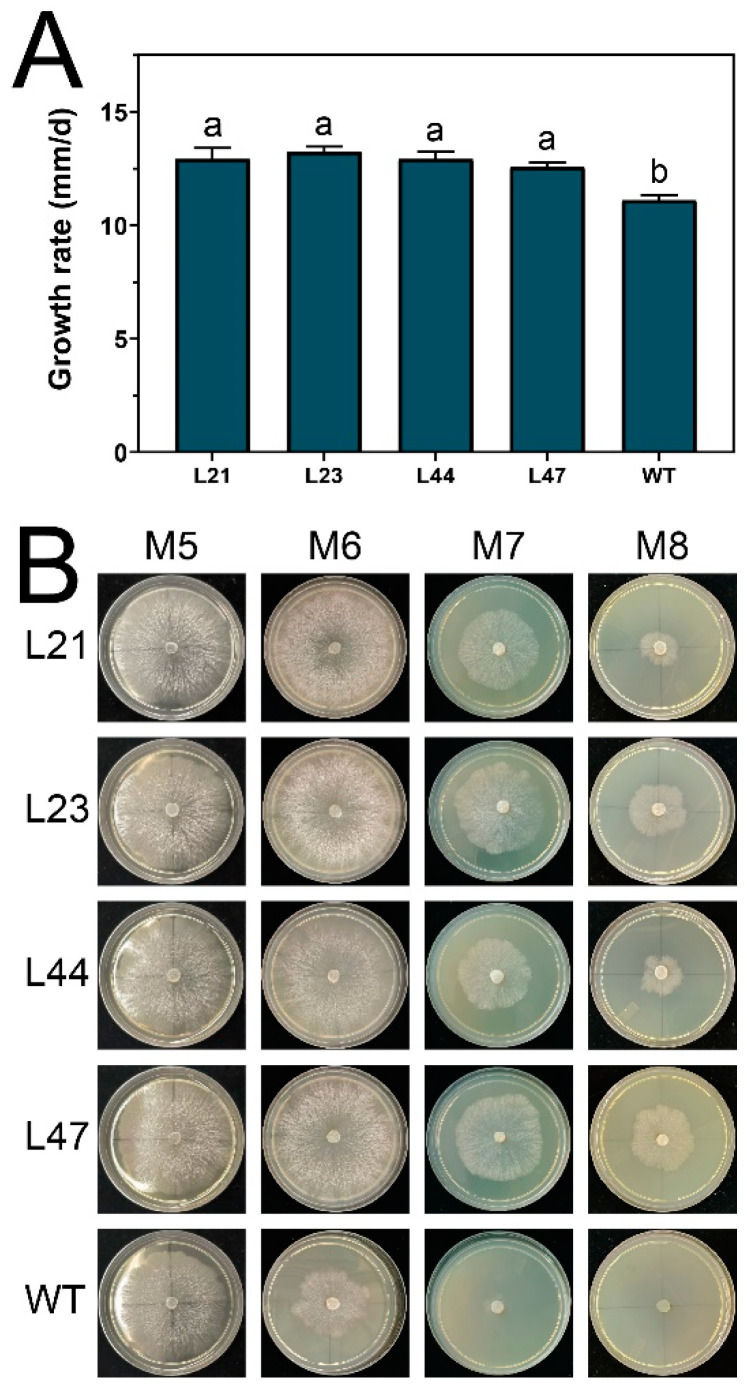
Growth rate of mycelium at 30 °C. The growth rate of mycelium at 30 °C (**A**) and the condition of mycelium after four successive generations of subculturing at 30 °C continuously (**B**). Values in a column with different letters are significantly different at *p* ≤ 0.05. The values presented are the means ± standard deviations (SDs) of three independent experiments.

**Figure 5 microorganisms-12-00518-f005:**
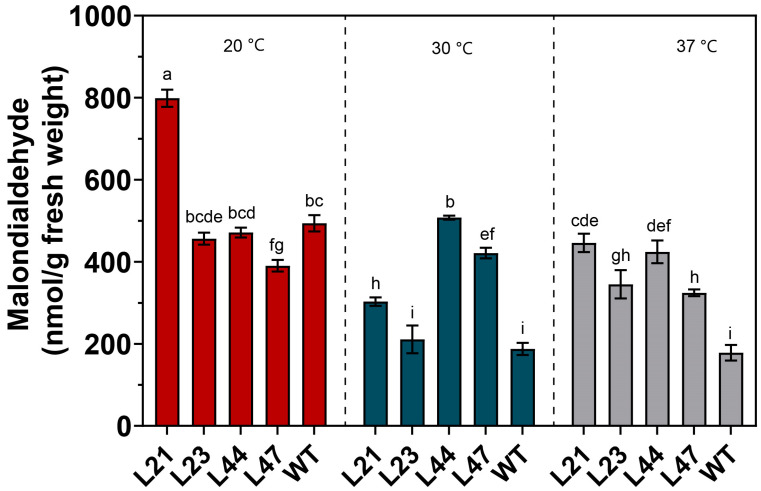
Content of malondialdehyde in mycelium. Values in a column with different letters are significantly different at *p* ≤ 0.05. The values presented are the means ± standard deviations (SDss) of three independent experiments.

**Figure 6 microorganisms-12-00518-f006:**
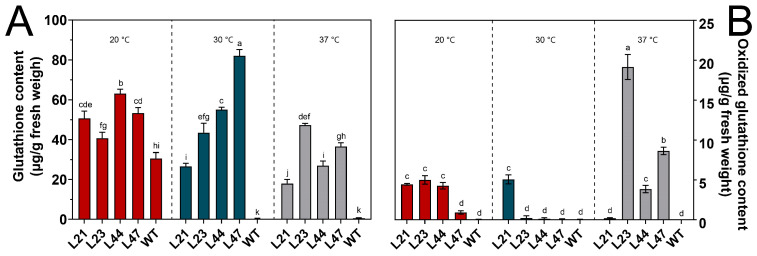
Content of GSH (**A**) and GSSG (**B**) in mycelium at different temperatures. The mean (±SD) was calculated from three replicates for each treatment. Values in a column with different letters are significantly different at *p* ≤ 0.05. The values presented are the means ± standard deviations (SDs) of three independent experiments.

**Figure 7 microorganisms-12-00518-f007:**
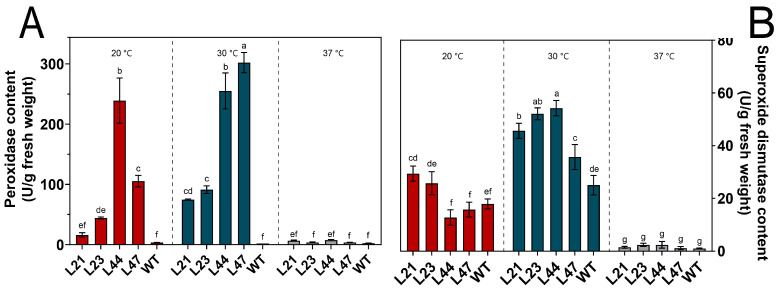
Activity of POD (**A**) and SOD (**B**) in mycelium at different temperatures. The mean (±SD) was calculated from three replicates for each treatment. Values in a column with different letters are significantly different at *p* ≤ 0.05. The values presented are the means ± standard deviations (SDs) of three independent experiments.

## Data Availability

No new data were created or analyzed in this study. Data sharing is not applicable to this article.

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
