# Peer review of "Enhanced Heat Resistance in Morchella eximia by Atmospheric and Room Temperature Plasma"

_microorganisms, 2024, doi:10.3390/microorganisms12030518_

Round 1

Reviewer 1 Report

Comments and Suggestions for Authors

The authors have mutagenized a strain of Morchella eximia by ATRP aiming at generating temperature tolerant strains. The method might indeed be useful to solve a problem in the outdoor cultivation of this organism.
The manuscript is, however, not always clear in the presentation of results and sometimes conclusions are drawn not supported by the data. Adapting the manuscript in several parts would improve the soundness of conclusions and the readability of the manuscript.

Line 2 (Title). The word “Breeding” is not appropriate in the title. Breeding is something different that just mutagenizing a strain. Breeding includes improving a strain for cultivation and the authors have not shown how the strain performs in producing fruiting bodies. The word “breeding” is also not used in the abstract.
Line 10: “..Mel-7”. The species name should also be mentioned in the abstract.
Lines 75-76: Therefore…cultivation.” I think that the present cultivation methods are outdoor cultivation. Under these conditions, no temperature control is possible. If that is true, this is an important reason to generate high temperature tolerant strains. You should mention this. I also wonder if the high temperature is especially a problem for mycelium growth or is it also a problem for the mushroom production? If the cultivation is done outdoor, is the mycelium growth period done in the spring/summer and mushroom production in the autumn? It might also be useful for readers unfamiliar to the cultivation of this mushroom to refer to a review (Liu et al. 2023. Large scale field cultivation of Morchella and relevance of basic knowledge for its steady production.
Line 128. Describe how the homogenizer was used (device brand, speed).
Line 229-232: Mention the sample replicate number.
Lines 247-248: Here, the authors state that the mycelium maintained its vitality up to a threshold of 43 o C. Figure 2B shows that the strain maintained its vitality somewhere between 39 and 41 degrees, and not up to 43 degrees. I see no differences between 41, 43, 45 and 47 degrees in growth of Mel-7. The authors should also explain in the legend what the red arrow indicates in Fig. 2B. In the legend of this figure also change “phenotype” into “growth”. Also add in the last sentence of the legend that the values refer to figure 2A.
Line 276: remove “even”.
Line 289: “During the cultivation process..”. I assume this is the mushroom cultivation process. Please clarify this.
Line 296: “..generation..”. Better: "..four generations, each having a duration of 3 days...". Why are the generations indicated as  M5 through M8?
Lines 296-297: "..as cultivation time increased.." Better: "with each generation.."
Lines 300-301: “This study….of 30 oC.” But figure 4B shows that the mutated strains are not stable since after 3 generations the growth is already reduced considerably.
Paragraph 3.4 Lipid peroxidation. Add here an explanation why you measure this. "Heat induces oxidative stress and causes lipid peroxidation of membranes. Malondialdehyde is one of the final products of membrane fatty acids peroxidation and commonly known as a marker of oxidative stress.”
Lines 323-324: A beneficial response to heat stress would be a reduction in membrane damage, i.e. reduction in fatty acids peroxidation and thus MDA concentrations compared to the wild strain. That is not the case. You should address this here or in the Discussion paragraph.
Paragraph 3.5 Non-enzymatic antioxidants. Add a short note why the concentrations of these compounds are indicative for stress. As for the MDA paragraph, I think the expression changes in GSH and GSSG expressed as % increase or decrease compared to previous temperature conditions is not that informative. The expression levels per se under different temperature conditions compared to the wild type is more informative.
Lines 441-442: “… practical utility …conditions.” But Figure 4 shows that there is a decrease in growth rate after 4 generation, thus not stable and not practical useful.
Lines 443-444: There experiments show a "survival" of a 48 h exposure to 37 oC, but not a improved growth at that temperature.
Lines 473-475: “Interestingly,…thermal response.” There is indeed a strain specific thermal response. But the conclusion should be that MDA is not indicative for temperature tolerance, since one would expect a degrees in MDA concentration compared to the control indicating less membrane damage.
Lines 476-483: I think the most important message here is that the GSH levels of the mutant strains remain relatively high whereas GSH level almost disappears in the wild strain. It is generally considered as healthy when GSGH levels are up.
Line 490: “redox homeostasis.” For an effective redox homeostasis of the wild strain I would expect that the ratio GSH/GSSG would be constant. That is not the case here.
Paragraph 4.4 AS for the GSG/GSG ratio's, the authors should point to the higher activities of POD and SOD compared to the wild strain, and not focus only on the change in the activities of these enzymes between temperatures. The higher activities indicate that in mutated strains these enzymes might indeed play a role in defence to higher temperatures.
Lines 528-530: This indicates that even for the mutated strains 37 oC is too high.
Line 544: “thrive in temperatures up to 37 oC.” Again, that is not true. I think the optimal temp for the mutants is 30 oC, underpinned also by the growth in figure 4B and the activity of ROS related enzyme activities.

Author Response

Dear editors and reviewers,

Subject: Manuscript ID: microorganisms-2809259, Title: Breeding thermo-tolerant stains of Morchella eximia by Atmospheric and Room Temperature Plasma

I hope this letter finds you well. I am writing to express my gratitude for the thorough review and constructive feedback provided by you and the esteemed reviewers on my manuscript, titled "Breeding thermo-tolerant stains of Morchella eximia by Atmospheric and Room Temperature Plasma," assigned the Manuscript ID: microorganisms-2809259.

I appreciate the time and effort invested in evaluating my work, and I am grateful for the valuable insights that have undoubtedly enhanced the quality of the manuscript. I have carefully considered all the comments and suggestions and have made the necessary revisions to address each concern raised during the review process.

In the revised manuscript, I have provided detailed responses to each comment, outlining the specific changes made in response to the reviewers' recommendations. Additionally, I have attached a marked-up copy of the manuscript to facilitate a quick comparison with the original submission.

I am confident that the revisions have strengthened the manuscript, aligning it more closely with the high standards of Microorganisms. I kindly request you to reevaluate the manuscript at your earliest convenience.

Comment 1: Line 2 (Title). The word “Breeding” is not appropriate in the title. Breeding is something different that just mutagenizing a strain. Breeding includes improving a strain for cultivation and the authors have not shown how the strain performs in producing fruiting bodies. The word “breeding” is also not used in the abstract.

Response 1: I have changed the title into “Enhanced heat resistance in Morchella eximia by Atmospheric and Room Temperature Plasma”. See in line 2.
Comment 2: Line 10: “..Mel-7”. The species name should also be mentioned in the abstract.

Response 2: I added the species name in front of “Mel-7”. See in line 10.
Comment 3: Lines 75-76: Therefore…cultivation.” I think that the present cultivation methods are outdoor cultivation. Under these conditions, no temperature control is possible. If that is true, this is an important reason to generate high temperature tolerant strains. You should mention this. I also wonder if the high temperature is especially a problem for mycelium growth or is it also a problem for the mushroom production? If the cultivation is done outdoor, is the mycelium growth period done in the spring/summer and mushroom production in the autumn? It might also be useful for readers unfamiliar to the cultivation of this mushroom to refer to a review (Liu et al. 2023. Large scale field cultivation of Morchella and relevance of basic knowledge for its steady production.

Response 3: That's correct. I added the reason for generating high-temperature strains at the end of that paragraph. See in ling 77-82. The cultivation of morel mushrooms typically takes place around October, with the harvesting period occurring from January to the end of March or early April of the following year. The specific harvesting period depends on the different varieties of morel mushrooms. High temperatures can affect the growth of morel mycelium, with the most significant impact occurring during the fruiting period. If the temperature becomes too high, there is a high risk of a complete crop failure.
Comment 4: Line 128. Describe how the homogenizer was used (device brand, speed).

Response 4: Done. See in line 134.
Comment 5: Line 229-232: Mention the sample replicate number.

Response 5: Done. See in line 235-236.
Comment 6: Lines 247-248: Here, the authors state that the mycelium maintained its vitality up to a threshold of 43 o C. Figure 2B shows that the strain maintained its vitality somewhere between 39 and 41 degrees, and not up to 43 degrees. I see no differences between 41, 43, 45 and 47 degrees in growth of Mel-7. The authors should also explain in the legend what the red arrow indicates in Fig. 2B. In the legend of this figure also change “phenotype” into “growth”. Also add in the last sentence of the legend that the values refer to figure 2A.

Response 6: In the text, I did mention" the mycelium maintained its vitality up to a threshold of 43°C." See in line 254-255. In Fig 2B, I have annotated the meaning of the red arrow. I have changed “phenotype” into “growth”. See in line 266 and 269.

I want to clarify that at 41°C, the mycelium can grow, as indicated by the red arrow. However, after being cultured at 43, 45, or 47°C for 1.5 hours, followed by one week of cultivation at 20°C, the mycelium did not grow. This indicates that 43°C is a critical temperature. If the mycelium cannot grow at 43°C, it certainly cannot grow at 45 or 47°C. Therefore, the mycelium at 43, 45, or 47°C behaves similarly.
Comment 7: Line 276: remove “even”.

Response 7:Done.
Comment 8: Line 289: “During the cultivation process..”. I assume this is the mushroom cultivation process. Please clarify this.

Response 8:The cultivation processes described here are all targeted at mycelial growth, and I have specifically added in the text that this is focused on mycelium.
Comment 9: Line 296: “..generation..”. Better: "..four generations, each having a duration of 3 days...". Why are the generations indicated as  M5 through M8?

Response 9: Morel mushrooms are typical ascomycetes. One typical characteristic of ascomycetes is that after multiple generations of cultivation, the mycelium may age or degenerate, leading to a decrease in vitality. In our experiments, we found that after more than 11 generations of continuous cultivation, most mutant strains' mycelium deteriorated, including the starting strain. Therefore, during our experiments, we found no significant differences in phenotype and strong vitality of mycelium between the M4 and M9 generations. Thus, for the consecutive high-temperature tolerance experiments, we chose the M5 to M8 generations.
Comment 10: Lines 296-297: "..as cultivation time increased.." Better: "with each generation.."

Response 10: Done.
Comment 11: Lines 300-301: “This study….of 30 oC.” But figure 4B shows that the mutated strains are not stable since after 3 generations the growth is already reduced considerably.

Response 11: The increased heat resistance observed in the mutated strains is compared to the starting strain WT. It is evident from Figure 4B that the wild-type WT mycelium growth rate sharply declines in the second generation (M6) of the heat tolerance experiment, and by the third generation (M7), it has almost ceased growing. In contrast, the mutated strains exhibit strong vitality in the second generation of the heat tolerance experiment, and although the growth rate decreases in the third generation, they still show some advantage over WT. By the fourth generation of the heat tolerance experiment, their mycelium continues to grow. This indicates that compared to WT, the mycelium of the mutated strains still possesses strong heat resistance.
Comment 12: Paragraph 3.4 Lipid peroxidation. Add here an explanation why you measure this. "Heat induces oxidative stress and causes lipid peroxidation of membranes. Malondialdehyde is one of the final products of membrane fatty acids peroxidation and commonly known as a marker of oxidative stress.”

Response 12: Done.
Comment 13: Lines 323-324: A beneficial response to heat stress would be a reduction in membrane damage, i.e. reduction in fatty acids peroxidation and thus MDA concentrations compared to the wild strain. That is not the case. You should address this here or in the Discussion paragraph.

Response 13:I discussed and explained. See in line483-484, 488-491.
Comment 14: Paragraph 3.5 Non-enzymatic antioxidants. Add a short note why the concentrations of these compounds are indicative for stress. As for the MDA paragraph, I think the expression changes in GSH and GSSG expressed as % increase or decrease compared to previous temperature conditions is not that informative. The expression levels per se under different temperature conditions compared to the wild type is more informative.

Response 14: Done. See in line 341-345, 347, 350-353,355-360.
Comment 15: Lines 441-442: “… practical utility …conditions.” But Figure 4 shows that there is a decrease in growth rate after 4 generation, thus not stable and not practical useful.

Response 15: The heat tolerance we are discussing here is relative. Compared with the wild type, the mutant strain does show stronger vitality after four consecutive generations at 30 degrees Celsius, indicating that the wild type does have some heat resistance. In actual production, extreme weather conditions do not persist for a long time. As shown in Figure 4, when cultured continuously for two generations at 30 degrees Celsius, the mutant strain exhibits strong vitality, while the vitality of the wild-type mycelium drops sharply.
Comment 16: Lines 443-444: There experiments show a "survival" of a 48 h exposure to 37 oC, but not a improved growth at that temperature.

Response 16: This statement compares the growth improvement to the wild type (WT). I will add a comment in the manuscript suggesting a comparison to the WT. See in line 467.
Comment 17: Lines 473-475: “Interestingly,…thermal response.” There is indeed a strain specific thermal response. But the conclusion should be that MDA is not indicative for temperature tolerance, since one would expect a degrees in MDA concentration compared to the control indicating less membrane damage.

Response 17:Done. See in line 505-507.
Comment 18: Lines 476-483: I think the most important message here is that the GSH levels of the mutant strains remain relatively high whereas GSH level almost disappears in the wild strain. It is generally considered as healthy when GSGH levels are up.

Response 18:Yes, we found in our experiments that the most significant difference between the mutant strains and the wild type is the levels of GSH and GSSG, and many studies suggest that an increase in GSH levels is considered healthy.
Comment 19: Line 490: “redox homeostasis.” For an effective redox homeostasis of the wild strain I would expect that the ratio GSH/GSSG would be constant. That is not the case here.

Response 19: I have rewritten this conclusion, suggesting that the main reason for the WT's intolerance to high temperatures may be its lower levels of GSH and GSSG. See in line 521-523.
Comment 20: Paragraph 4.4 AS for the GSG/GSG ratio's, the authors should point to the higher activities of POD and SOD compared to the wild strain, and not focus only on the change in the activities of these enzymes between temperatures. The higher activities indicate that in mutated strains these enzymes might indeed play a role in defence to higher temperatures.

Response 20: I discussed in the paper the enhanced role of elevated SOD and POD in the mutant strains in heat tolerance. See in line 566-57.
Comment 21: Lines 528-530: This indicates that even for the mutated strains 37 oC is too high.

Response 21: I did not find this information in lines 528-530.
Comment 22: Line 544: “thrive in temperatures up to 37 oC.” Again, that is not true. I think the optimal temp for the mutants is 30 oC, underpinned also by the growth in figure 4B and the activity of ROS related

Response 22: Yes, in this study, the mutant strains showed some tolerance at 30°C.

Reviewer 2 Report

Comments and Suggestions for Authors

In this study, the author optimized the mutation process of Mel-7 mycelium by ARTP, and studied the thermal adaptability and physiological response of the mutated strains. The results showed a significant relationship between the duration of ARTP induced mutagenic exposure and mortality rate, with an exposure time of 40 seconds and 43 ℃ as the ideal screening temperature. The strains L21, L23, L44, and L47 showed good growth and high temperature resistance under 30℃ conditions. This study provides a new explanation for the thermal adaptability of Mel-7 mycelium and an effective method for identifying mutant strains, providing valuable insights for academic and industrial purposes.

1. The English of the manuscript needs to be corrected, preferably by a native speaker. There are a lot of awkward sentences.

2. In line 29 of the Introduction, (Morchella spp.) needs to be italicized.

3. 3.3 The heat tolerance of mutant strains was only studied at 30°C. In view of the physiological and biochemical reactions later in the study, the relevant data under the growth conditions of 37°C were studied. Whether the growth of mutant strains under growth conditions above 30°C has been tested, it is recommended to supplement them

4. When studying the physiology and biochemistry of mutant strains, the physiological and biochemical conditions of the strains at 20℃, 30℃ and 37℃ were tested, respectively. Please add why only 1 d of heat treatment was applied at 37 ℃ instead of continuous incubation.

Comments on the Quality of English Language

Extensive editing of English language required

Author Response

Dear editors and reviewers,

Subject: Manuscript ID: microorganisms-2809259, Title: Breeding thermo-tolerant stains of Morchella eximia by Atmospheric and Room Temperature Plasma

I hope this letter finds you well. I am writing to express my gratitude for the thorough review and constructive feedback provided by you and the esteemed reviewers on my manuscript, titled "Breeding thermo-tolerant stains of Morchella eximia by Atmospheric and Room Temperature Plasma," assigned the Manuscript ID: microorganisms-2809259.

I appreciate the time and effort invested in evaluating my work, and I am grateful for the valuable insights that have undoubtedly enhanced the quality of the manuscript. I have carefully considered all the comments and suggestions and have made the necessary revisions to address each concern raised during the review process.

In the revised manuscript, I have provided detailed responses to each comment, outlining the specific changes made in response to the reviewers' recommendations. Additionally, I have attached a marked-up copy of the manuscript to facilitate a quick comparison with the original submission.

I am confident that the revisions have strengthened the manuscript, aligning it more closely with the high standards of Microorganisms. I kindly request you to reevaluate the manuscript at your earliest convenience.

Comment 1:The English of the manuscript needs to be corrected, preferably by a native speaker. There are a lot of awkward sentences.

Response 1: We have submitted the manuscript to a professional English editing service for language polishing.

Comment 2: In line 29 of the Introduction, (Morchella spp.) needs to be italicized.

Response 2: Done.

Comment 3: 3.3 The heat tolerance of mutant strains was only studied at 30°C. In view of the physiological and biochemical reactions later in the study, the relevant data under the growth conditions of 37°C were studied. Whether the growth of mutant strains under growth conditions above 30°C has been tested, it is recommended to supplement them

Response 3: In the study, four heat-resistant strains were selected through screening at 37°C as shown in Fig. 3 for further research, indicating their heat resistance at 37°C. In the subsequent study, physiological and biochemical characteristics were measured at both 30°C and 37°C.

Comment 4: When studying the physiology and biochemistry of mutant strains, the physiological and biochemical conditions of the strains at 20℃, 30℃ and 37℃ were tested, respectively. Please add why only 1 d of heat treatment was applied at 37 ℃ instead of continuous incubation.

Response 4: Edible fungi are a type of mesophilic fungus, with morel mushrooms cultivated at even lower temperatures, not exceeding 25 ℃. In this study, the mycelium was unable to grow when cultured at 37 ℃. When the mycelium was cultured at 37 ℃ for more than 72 hours and then returned to 20 â„ƒ for continued cultivation, the fungal growth ceased. Therefore, the mycelium cannot be continuously cultured at 37 ℃.

Reviewer 3 Report

Comments and Suggestions for Authors

I enjoyed reading this paper about using ARTP to improve the thermal tolerance of a morel mushroom. The authors did a great job designing the experiments and presenting the data. The writing and figures are easy to follow, helping the readers navigate different concepts smoothly. I believe the manuscript is in pretty good shape but could use a few edits to improve its clarity.

1.        Could the authors provide more details about the background of the mycelium materials? Is it monokaryotic or dikaryotic? I suppose monokaryotic mycelium would be more suitable for mutagenesis experiments. 

2.        Page 3, Line 145: what is PBD?

3.        Figure 2B and C: I believe white would be better than red for the temperature labels. Please also consider adding scale bars to all figure panels with photos.

4.        Page 6, Line 248: I had a hard time seeing the difference between the photo at 43C versus the ones at 45 and 47C. Actually, it seems like 41C would be a better cutoff than 43C. Could the authors highlight the differences by adding measurements or labels to the photos?

5.        Figure 3A: how was the growth rate measured?

6.        Figure 3-7: all the bar graphs have some alphabetical labels that seem to mean nothing. Could we remove them?

Author Response

Dear editors and reviewers,

Subject: Manuscript ID: microorganisms-2809259, Title: Breeding thermo-tolerant stains of Morchella eximia by Atmospheric and Room Temperature Plasma

I hope this letter finds you well. I am writing to express my gratitude for the thorough review and constructive feedback provided by you and the esteemed reviewers on my manuscript, titled "Breeding thermo-tolerant stains of Morchella eximia by Atmospheric and Room Temperature Plasma," assigned the Manuscript ID: microorganisms-2809259.

I appreciate the time and effort invested in evaluating my work, and I am grateful for the valuable insights that have undoubtedly enhanced the quality of the manuscript. I have carefully considered all the comments and suggestions and have made the necessary revisions to address each concern raised during the review process.

In the revised manuscript, I have provided detailed responses to each comment, outlining the specific changes made in response to the reviewers' recommendations. Additionally, I have attached a marked-up copy of the manuscript to facilitate a quick comparison with the original submission.

I am confident that the revisions have strengthened the manuscript, aligning it more closely with the high standards of Microorganisms. I kindly request you to reevaluate the manuscript at your earliest convenience.

Comment 1: Could the authors provide more details about the background of the mycelium materials? Is it monokaryotic or dikaryotic? I suppose monokaryotic mycelium would be more suitable for mutagenesis experiments.

Response 1: The material used in this study is dikaryotic mycelium.

Comment 2: Page 3, Line 145: what is PBD?

Response 2: This is my mistake, it should be PDB.

Comment 3:Figure 2B and C: I believe white would be better than red for the temperature labels. Please also consider adding scale bars to all figure panels with photos.

Response 3: Done.

Comment 4: Page 6, Line 248: I had a hard time seeing the difference between the photo at 43C versus the ones at 45 and 47C. Actually, it seems like 41C would be a better cutoff than 43C. Could the authors highlight the differences by adding measurements or labels to the photos?

Response 4: We added a clarification to Fig.2 using white arrows. We are trying to determine the highest temperature (lowest lethal temperature) at which the mycelium does not grow. After heat treatment at 41°C for 1.5 hours and then incubation at 20°C for 7 days, the mycelium continued to grow. However, after incubation at 43°C for 1.5 hours and then incubation at 20°C for 7 days, the mycelium did not grow. This indicates that 43°C is the lowest lethal temperature.

Comment 5: Figure 3A: how was the growth rate measured?

Response 5: Done.See Paragraph 2.4.

Comment 6:Figure 3-7: all the bar graphs have some alphabetical labels that seem to mean nothing. Could we remove them?

Response 6: User believes these letters should clearly indicate the differences between each pair of bars. Bars with the same letter do not show significant differences, while bars with different letters indicate significant differences. Therefore, I believe these letters should not be removed.